# Foundation of Scalable Constraint Learning from Human Feedback

## Abstract

Constraint learning from human feedback (CLHF) has garnered significant interest in the domain of safe reinforcement learning (RL) due to the challenges associated with designing constraints that elicit desired behaviors. However, a comprehensive theoretical analysis of CLHF is still missing. This paper addresses this gap by establishing a theoretical foundation. Concretely, trajectory-wise feedback, which is the most natural form of feedback, is shown to be helpful only for learning chance constraints. Building on this insight, we propose and theoretically analyze algorithms for CLHF and for solving chance constrained RL problems. Our algorithm is empirically shown to outperform an existing algorithm.

## 1 Introduction

Designing an appropriate reward function for reinforcement learning (RL) is a challenging task, as a poorly designed reward function can lead to various unintended behaviors (Krakovna et al., 2016). One approach to mitigate this issue is inverse reinforcement learning (IRL), where the reward function is inferred from expert demonstrations (Ng and Russell, 2000; Ziebart et al., 2008; 2010). However, acquiring expert demonstrations can be costly or even infeasible in certain situations. As a result, there has been growing interest in learning the reward function from human feedback (Wirth et al., 2017), particularly using preference datasets for fine-tuning large language models (Ouyang et al., 2022).

Constraint learning from human feedback (CLHF) has attracted considerable attention (Scobee and Sastry, 2020; Glazier et al., 2021; Papadimitriou and Anwar, 2021; Stocking et al., 2021; Malik et al., 2021; Poletti et al., 2023; Lindner et al., 2024; Zhu et al., 2024; Dai et al., 2024) due to the crucial importance of safety in real-world applications and the inherent challenges in designing cost functions (Krakovna et al., 2016). Nonetheless, a comprehensive theoretical analysis of CLHF is still missing. This paper fill this gap by establishing a theoretical foundation for CLHF, with emphasis on scalability to flexible models such as neural networks, ability to handle multiple constraints and ambiguity in human feedback.

Concretely, in Section 3, we investigate what form of human feedback is appropriate for what type of constraints. In particular, we show that trajectory feedback, which will be explained later and is thought to be a most natural form of feedback, is helpful to solve only chance-constrained RL problems (Chow et al., 2017).

In Section 4, we propose a new learning scheme for CLHF that can handle multiple constraints and ambiguity in human feedback. We theoretically analyze it and show that the scheme is provably able to learn a component necessary to solve chance-constrained RL problems.

In Section 5, we propose a new policy gradient algorithm for solving chance-constrained RL problems. It is an extension of Chen et al. (2024) to problems with a more general chance constraints.

In Section 6, we empirical validated the superior performance of our algorithm in both tabular and function approximation cases, confirming that our algorithm scales well.

## 2 BACKGROUND AND NOTATIONS

We denote sets by curly alphabets with some exceptions to follow mathematical convention, such as the set of natural numbers, $\mathbb{N} := \{1, 2, 3, \ldots\}$, and the set of reals, $\mathbb{R}$. The set of integers from 1 to $N$ is denoted by $[N]$. For a finite set $\mathcal{A}$, the set of all probability distributions over $\mathcal{A}$ is denoted by $\Delta(\mathcal{A})$. The indicator function is denoted by $\mathbb{I}(C)$, which returns 1 if $C$ is true and 0 otherwise.

### 2.1 CONSTRAINED MARKOV DECISION PROCESSES (CMDPs)

RL problems are often formulated as Markov Decision Processes (MDPs). [1]

**Definition 1** (MDP). *The MDP is defined by six elements (Puterman, 1994), the (finite) state space $\mathcal{S}$, the (finite) action space $\mathcal{A}$, horizon $H$, the initial state distribution $P_1 \in \Delta(\mathcal{S})$, state-transition dynamics $P : \mathcal{S} \times \mathcal{A} \to \Delta(\mathcal{S})$, and the reward function $r : \mathcal{S} \times \mathcal{A} \times \mathcal{S} \to [-1, 1]$.*

A policy is a mapping from $[H] \times \mathcal{S}$ to $\Delta(\mathcal{A})$. Given an MDP, a learner aims at finding an optimal policy, which has the highest return, as defined below. We let $\mathbb{E}^\pi$ mean the expectation under a policy $\pi$, *i.e.*, $A_h \sim \pi_h(\cdot|S_h)$. We use $\mathbb{P}^\pi$ to similarly mean a probability distribution over the space of trajectories under the policy $\pi$.

**Definition 2** (Value Function and Return). *For any policy $\pi$, time step $h$, and a real-valued function $f$, let $v_{f,h}^\pi : \mathcal{S} \to \mathbb{R}$ be defined as a function*

$$v_{f,h}^\pi(s) = \mathbb{E}^\pi \left[ \sum_{t=h}^{H} f(S_t, A_t, S_{t+1}) \,\middle|\, S_h = s \right],$$

*which is called the value function if $f = r$. The return is defined as $v_{f,1}^\pi(P_1) := \sum_{s \in \mathcal{S}} v_{f,1}^\pi(s) P_1(s)$, and the optimal policy is defined as a maximizer of the return with respect to $\pi$.*

A trajectory is a sequence $(s_1, a_1, \ldots, s_H, a_H, s_{H+1})$, and its random version is denoted as $T$. A state-action-state triplet $(s, a, s')$ is said to be valid if $P(s'|s, a) > 0$. Similarly a trajectory is valid if it has non-zero probability under some policy. We let $\mathcal{T}$ be the set of all valid trajectories.

We consider two problem formulations of constrained RL problems. We call the first one as chance-CMDP (C-CMDP), which is based on constraint violation probability (Chow et al., 2017) and defined as follows. We call the type of constraints used in the C-CMDP as chance constraints.

**Definition 3** (C-CMDP). *Suppose an MDP, a positive integer $N$, cost functions $\{c_n|\mathcal{S} \times \mathcal{A} \times \mathcal{S} \to [-1, 1], n \in [N]\}$, and a scalar $\delta \in (0, 1)$, which determines an admissible constraint violation probability. In the C-CMDP induced by the MDP and cost functions, the learner aims at solving the following constrained optimization problem:*

$$\max_\pi v_1^\pi(P_1) \ \text{s.t.} \ \mathbb{P}^\pi(\nu(T) \le 0) \ge 1 - \delta, \ \text{where} \ \nu(\tau) := \max_{n \in [N]} \sum_{h \in [H]} c_n(s_h, a_h, s_{h+1}). \tag{1}$$

The second one is expected-CMDP (E-CMDP), which is based on cumulative cost (Altman, 1999) and defined as follows. We call the type of constraints used in the E-CMDP as expected constraints.

**Definition 4** (E-CMDP). *Suppose an MDP, a positive integer $N$, and cost functions $\{c_n|\mathcal{S} \times \mathcal{A} \times \mathcal{S} \to [-1, 1], n \in [N]\}$. In the E-CMDP induced by the MDP and cost functions, the learner aims at solving the following constrained optimization problem:*

$$\max_\pi v_{r,1}^\pi(P_1) \ \text{s.t.} \ \nu'(\pi) \le 0, \ \text{where} \ \nu'(\pi) := \max_{n \in [N]} v_{c_n,1}^\pi(P_1).$$

---

[1]While time-independent reward and dynamics are considered for simplicity, it is straightforward to extend all results and algorithms to a time-dependent reward and dynamics setting. Also, the finite MDP setting is considered to avoid distraction due to complicated measure theoretic arguments. We believe most of results can be extended to the continuous MDP setting.

## 2.2 Constraint Learning from Human Feedback

We consider a setting where the underlying problem is a constrained RL (either C-CMDP or E-CMDP), but *cost functions $(c_n)_{n=1}^N$ need to be inferred from a dataset containing human feedback.*[2] We suppose the following types of feedback and data collection process (c.f. Figure 1):

- **Trajectory feedback:** a given dataset $\mathcal{D} = \{(\tau_k, y_k)|k = 1, 2, \ldots\}$ consists of a trajectory $\tau_k$ paired with feedback $y_k \in \{-1, +1\}$. The $k$-th trajectory is assumed to be sampled from a probability distribution that may depend on all previous trajectories and feedback, and the $k$-th feedback is sampled from a fixed conditional probability distribution $\rho(\cdot|\tau_k)$.

- **Policy feedback:** a given dataset $\mathcal{D} = \{(\pi_k, y_k)|k = 1, 2, \ldots\}$ consists of a policy $\pi_k$ paired with a feedback $y_k \in \{-1, +1\}$. The $k$-th policy is assumed to be sampled from a probability distribution that may depend on all previous policies and feedback, and the $k$-th feedback is sampled from a fixed conditional probability distribution $\rho'(\cdot|\pi_k)$.

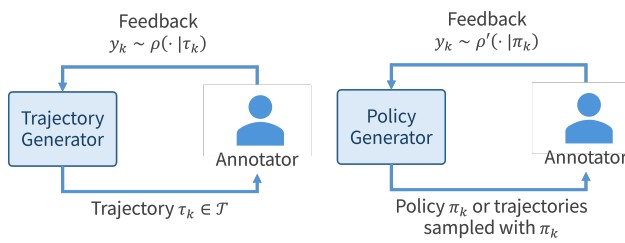

Figure 1: Data collection process. The trajectory (resp. policy) generator may choose a trajectory (resp. policy) based on all previous trajectories (policies) and feedback, and feedback is sampled from a fixed conditional probability distribution. There may be more than single annotator as long as the feedback distribution is fixed.

We believe that trajectory feedback is less costly to collect since an annotator just need to look at a single trajectory and give feedback to it. In contrast, policy feedback requires an annotator to look at multiple trajectories from a single policy and determine if the policy satisfies constraints. Maybe an exceptional situation is where expert policies are accessible as in, e.g., Lindner et al. (2024).

Since CLHF is clearly impossible if human feedback is generated in an arbitrary manner, we often impose the following mild assumption.

**Assumption 1.** *For any trajectories $\tau$ and $\tau'$, $\rho(+1|\tau) \geq \rho(+1|\tau')$ if and only if $\nu(\tau) \geq \nu(\tau')$, and $\rho(+1|\tau) = 0.5$ if and only if $\nu(\tau) = 0$. Furthermore, $\rho'$ satisfies similar conditions but with policies instead of trajectories.*

## 3 Which Feedback Should Be Used When?

A natural question is which feedback is appropriate to what situation. This section is devoted to answer the question, and Table 1 summarizes results.

Table 1: Summary of Possibility and Impossibility Results.

| Feedback Type | Expected Constraint | Chance Constraint |
|---|---|---|
| TRAJECTORY FEEDBACK | IMPOSSIBLE | POSSIBLE |
| POLICY FEEDBACK | POSSIBLE | IMPOSSIBLE |

A formal version of the following impossibility result is provided and proven in Appendix A.

**Proposition 1** (Informal). *Cost functions learned through trajectory (resp. policy) feedback may induce an E-CMDP (resp. C-CMDP) whose safe policy is not safe in the true E-CMDP (resp. C-CMDP) induced by the true cost functions.*

Conversely, is it possible to learn meaningful constraints from trajectory (resp. policy) feedback for a C-CMDP (resp. E-CMDP)? The following possibility result answers it affirmatively.

---

[2]While $r$ is assumed to be known, our algorithm does not require it.

**Proposition 2.** *Suppose access to an oracle that maps a trajectory $\tau$ (resp. policy $\pi$) to $\mathbb{I}(2\rho(+1|\tau) - 1 > 0)$ (resp. $\mathbb{I}(2\rho'(+1|\pi) - 1 > 0)$). Then, under Assumption 1, it is possible to determine if a policy $\pi \in \Pi$ is safe or not for a C-CMDP (resp. E-CMDP) given $P_1$, $P$, and $r$.*

This result is obvious for the policy feedback case. In case of the trajectory feedback case, note that

$$\mathbb{P}^\pi(\nu(T) > 0) = \sum_{\tau \in \mathcal{T}} P_1(s_1) \prod_{h=1}^{H} P(s_{s+1}|s_h, a_h)\pi_h(a_h|s_h)\mathbb{I}(2\rho(+1|\tau) - 1 > 0).$$

Hence, given $P_1$, $P$, and $r$, it is possible to determine if a policy $\pi \in \Pi$ is safe or not with an oracle.

Propositions 1 and 2 together show that trajectory (resp. policy) feedback should be collected when one wants to use chance (resp. expected) constraints. Based on this observation and our belief stated before that trajectory feedback is less costly to collect, we focus on CLHF from trajectory feedback hereafter. Furthermore, the possibility result shows that an oracle in the statement suffices to find a near-optimal policy. The focus of the next section is how to estimate such an oracle.

## 4 Principled and Scalable CLHF

One approach for estimating an oracle is to learn a decision function $d : \mathcal{T} \to \mathbb{R}$ (Hastie et al., 2001). Concretely, let $\mathcal{D} := \{(\tau, y)\}$ be a set of trajectory-feedback pairs, and $\ell : \mathbb{R} \to [0, \infty)$ be a margin-based binary loss function. Then, an oracle is found by minimizing the empirical risk $\mathbb{E}_{(\tau,y)\sim\mathcal{D}}[\ell(yd(\tau))]$ with respect to $d$. Indeed, the minimizer $d^\star$ of the true risk, $\mathbb{E}_{(T,Y)}[\ell(Yd(T))]$, satisfies $\mathrm{sign}(d^\star(\tau)) = \mathrm{sign}(2\rho(1|\tau) - 1)$ under some technical conditions (Bartlett et al., 2003).

However, using functions over $\mathcal{T}$ is likely to be statistically inefficient as the structure of constraints is completely ignored. (See Section 2.) In order to leverage the structure of constraints for efficient learning, we propose to use the decision function explained below.

### 4.1 Proposed Decision Function

If $N$ and $(c_n)_{n=1}^N$ were all known, $d(\tau) = \max_{n\in[N]} d_n(\tau) = \max_{n\in[N]} \sum_{h\in[H]} c_n(s_h, a_h, s_{h+1})$, is a reasonable choice. Thus, we propose to use the following decision function:

$$d_{\mathbf{w}}(\tau) := \max_{m\in[M]} \sum_{h=1}^{H} \widehat{c}_{\mathbf{w},m}(s_h, a_h, s_{h+1}),$$

where $M$ is some positive integer treated as a hyper-parameter, $\widehat{c}_{\mathbf{w}} : \mathcal{S} \times \mathcal{A} \times \mathcal{S} \to [-H, H]^M$ is some differentiable parameterized model with a parameter vector $\mathbf{w}$, and $\widehat{c}_{\mathbf{w},m}$ denotes its $m$-th dimension of the output. Given a dataset $\mathcal{D}$, $\mathbf{w}$ is found by minimizing the empirical risk,

$$\mathfrak{R}_\mathcal{D}(\mathbf{w}) := \mathbb{E}_{(T,Y)\sim\mathcal{D}}[\ell(Yd_{\mathbf{w}}(T))], \tag{2}$$

hoping that it is a good estimate of the risk $\mathfrak{R}(\mathbf{w}) := \mathbb{E}_{(T,Y)}[\ell(Yd_{\mathbf{w}}(T))]$. In the sequel, we analyze the proposed decision function and how to choose $\ell$.

**Notations and Assumption:** Let $\mathcal{W} \subset \mathbb{R}^D$ and $\mathcal{H} := \{d_{\mathbf{w}}|\mathbf{w} \in \mathcal{W}, d_{\mathbf{w}}$ is differentiable at any $\mathbf{w}\}$ be the parameter space and a hypothesis class, respectively. For any $\alpha \in (0, \infty)$, the $\alpha$-covering number of $\mathcal{H}$ with the sup norm, $\|\cdot\|_\infty$, is denoted by $\mathcal{N}_\alpha$.

We also need the following technical assumption. It is satisfied by the logistic loss and squared loss.

**Assumption 2.** *The loss function $\ell$ has derivative $\ell'$, is $L$-Lipschitz continuous, and $\sigma$-strongly convex over $[-H, H]$, that is, $\ell(x) - (x - x')\ell'(x') - \sigma(x - x')^2 \geq \ell(x')$ for any $x, x' \in [-H, H]$.*

### 4.2 Theoretical Analysis

The following theorem, proven in Appendix B, shows that the minimization of Equation (2) leads to a reasonable estimate of the true decision function. It also shows how the number of data and its quality is related to the accuracy of the estimate.

**Theorem 1.** *Let $K := |\mathcal{D}|$. For any $\delta \in (0, 1)$, and $\alpha \in (0, \infty)$,*

$$\mathbb{P}\left(\forall(d, K) \in \mathcal{H} \times \mathbb{N}, \Re_{\mathcal{D}}(d^\star) - \Re_{\mathcal{D}}(d) \le \alpha L + \frac{L^2}{\sigma K} \log \frac{\mathcal{N}_\alpha}{\delta} - \frac{\sigma L_{2,K}(d, d^\star)}{2}\right) \ge 1 - \delta,$$

*where $L_{2,K}(d, d^\star) := \sum_{k=1}^{K}(d(\tau_k) - d^\star(\tau_k))^2/K$, and $d^\star : \mathcal{T} \to \mathbb{R}$ is the optimal decision function defined as*

$$d^\star(\tau) = \underset{x \in [-H, H]}{\arg\min} \, \mathbb{E}[\ell(Yx) \mid T = \tau].$$

In order to understand Theorem 1, let us consider a special case in which $\mathcal{H}$ is the set of all mappings from $\mathcal{S} \times \mathcal{A} \times \mathcal{S} \to [-H, H]$. Then, because its $\alpha$-covering number is $(2H/\alpha)^{|\mathcal{S} \times \mathcal{A} \times \mathcal{S}|}$,

$$0 \le \Re_{\mathcal{D}}(d^\star) - \Re_{\mathcal{D}}(\widehat{d^\star}) \le \frac{L^2|\mathcal{S} \times \mathcal{A} \times \mathcal{S}|}{\sigma K} \log \frac{2\sigma H L}{\delta L|\mathcal{S} \times \mathcal{A} \times \mathcal{S}|} - \frac{\sigma L_{2,K}(\widehat{d^\star}, d^\star)}{2},$$

where $\widehat{d^\star}$ is the minimizer of $\Re_{\mathcal{D}}(d)$, and we simplified and optimized the upper bound with respect to $\alpha$. Therefore, for any $K$,

$$d^\star \in \left\{d \,\middle|\, L_{2,K}(\widehat{d^\star}, d) \le \frac{2L^2|\mathcal{S} \times \mathcal{A} \times \mathcal{S}|}{\sigma^2 K} \log \frac{2\sigma H L}{\delta L|\mathcal{S} \times \mathcal{A} \times \mathcal{S}|}, d \in \mathcal{H}\right\}$$

with probability at least $1 - \delta$. Therefore, if sampled trajectories are diverse enough, $\widehat{d^\star}$ gets closer and closer to $d^\star$ as the dataset size increase. Interestingly, $\sigma L_{2,K}(\widehat{d^\star}, d^\star)$ is upper-bounded by $|\mathcal{S} \times \mathcal{A} \times \mathcal{S}|$ while it would be only upper-bounded by $|\mathcal{T}| \approx |\mathcal{S} \times \mathcal{A} \times \mathcal{S}|^H$, highlighting the importance of using our proposed decision function.

**Remark 1.** *Theorem 1 even holds for the active exploration setting, in which the learner is allowed to generate trajectories to reduce the uncertainty on $d^\star$ while being safe. To this end, one needs a pessimistic estimate of $d^\star$, but due to the shape of the decision function, existing count-based methods do not seem to be applicable. We leave active exploration as a future research direction.*

## 5 PRACTICAL POLICY GRADIENT ALGORITHM FOR C-CMDPs

Next, we discuss how to solve Equation (1) given access to cost functions. To begin with, we rewrite Equation (1) with Lagrangian as follows:

$$\max_{\pi} \min_{\lambda \ge 0} \left\{ v_1^\pi(P_1) + \lambda \left[\delta - \mathbb{P}^\pi\left(\max_{n \in [N]} \sum_{h=1}^{H} c_n(S_h, A_h, S_{h+1}) > 0\right)\right]\right\}. \tag{3}$$

While it is unclear if the strong duality holds, we propose to solve it using a primal-dual method as in (Chen et al., 2024). Since solving chance-constrained RL problems is still under active research, proposing and analyzing a new algorithm for chance-constrained RL problems goes beyond the scope of our paper.

To solve Equation (3) using a primal-dual method, we need its gradient with respect to policy parameters. Suppose a policy class, $\{\pi_{\mathbf{v}} | \mathbf{v} \in \mathbb{R}^D\}$, where $\mathbf{v}$ is a parameter vector. The following theorem provides the gradient expression of $\mathbb{P}^{\pi_{\mathbf{v}}}(\cdots)$ with respect to $\mathbf{v}$. Its proof is in Appendix C.

**Theorem 2.** *Let $C_{n,h} := \sum_{h'=1}^{h-1} c_n(S_{h'}, A_{h'}, S_{h'+1})$, $C_h := (C_{1,h}, \ldots, C_{N,h})$, and*

$$P_h^{\pi_{\mathbf{v}}}(s, a, c) := \mathbb{P}^{\pi_{\mathbf{v}}}\left(\max_{n \in [N]} C_{n,H+1} > 0 \,\middle|\, S_h = s, A_h = a, C_h = c\right).$$

*We have that*

$$\nabla_{\mathbf{v}} \mathbb{P}^{\pi_{\mathbf{v}}}\left(\max_{n \in [N]} C_{n,H+1} > 0\right) = \sum_{h=1}^{H} \mathbb{E}^{\pi_{\mathbf{v}}}[P_h^{\pi_{\mathbf{v}}}(S_h, A_h, C_h) \nabla_{\mathbf{v}} \ln \pi_{\mathbf{v},h}(A_h|S_h)].$$

---

**Algorithm 1:** CCPG

---

**Input:** Cost functions $(c_n)_{n=1}^N$, thresholds $(\theta_n)_{n=1}^N$, admissible violation probability $\delta \in [0, 1)$,
  mini-batch size $N_B$, learning rate $\alpha$.

1 Initialize $\eta$ to 0, and neural network parameters $\mathbf{w}_q$, $\mathbf{w}_P$, and $\mathbf{w}_\pi$ to arbitrary vectors;

2 **for** $k$ **from** 1 **to** $K$ **do**

3    **for** $n_B$ **from** 1 **to** $N_B$ **do**

4       Let $C_{n,1} = 0$ for all $n \in [N]$, reset an environment, and observe an initial state $s_1$;

5       **for** $h$ **from** 1 **to** $H$ **do**

6          Sample and execute an action $a_h \sim \pi_{\mathbf{w}_\pi,h}(\cdot|s_h, c_h)$;

7          Receive a reward $r_h$ and observe a next state $s_{h+1}$;

8          Let $C_{n,h+1} = C_{n,h} + c_n(s_h, a_h, s_{h+1})$ for all $n \in [N]$;

9       **end**

10       Let $P = \mathbb{I}(\sum_{h=1}^H c_{n,h} > \theta_n$ for some $n \in [N])$ and $R_h = \sum_{h'=h}^H r_{h'}$ for any $h \in [H]$;

11       **for** $h$ **from** $H$ **to** 1 **do**

12          Let $C_h = (C_{1,h}, \ldots, C_{N,h}) \in \mathbb{R}^N$ and $\lambda = \mathrm{softplus}(\eta)$;

13          $\Delta_q \leftarrow \Delta_q + \nabla_{\mathbf{w}_q}(q_{\mathbf{w}_q,h}(s_h, a_h, C_h) - R_h)^2$;

14          $\Delta_P \leftarrow \Delta_P + \nabla_{\mathbf{w}_P}(-P \log P_{\mathbf{w}_P,h}(s_h, a_h, C_h) - (1-P) \log(1 - P_{\mathbf{w}_P,h}(s_h, a_h, C_h))$;

15          $\Delta_\pi \leftarrow \Delta_\pi + \left(q_{\mathbf{w}_q,h}(s_h, a_h, C_h) - \lambda P_{\mathbf{w}_P,h}(s_h, a_h, C_h)\right)\nabla_{\mathbf{w}_\pi} \ln \pi_{\mathbf{w}_\pi,h}(a_h|s_h, C_h)$;

16       **end**

17       $\Delta_\eta \leftarrow \Delta_\eta + (\delta - P)\nabla_\eta \mathrm{softplus}(\eta)$;

18    **end**

19    $\mathbf{w}_q \leftarrow \mathbf{w}_q - \dfrac{\alpha \Delta_q}{HN_B}, \mathbf{w}_P \leftarrow \mathbf{w}_P - \dfrac{\alpha \Delta_P}{HN_B}, \mathbf{w}_\pi \leftarrow \mathbf{w}_\pi + \dfrac{\alpha \Delta_\pi}{N_B}$, and $\eta \leftarrow \eta - \alpha \Delta_\eta$;

20 **end**

21 **return** *Optimzied policy parameters* $\mathbf{w}_\pi$;

---

**Remark 2.** *Similarly to the original policy gradient theorem, Theorem 2 holds even if $P_h^{\pi_\mathbf{v}}$ in the gradient expression is replaced by $P_h^{\pi_\mathbf{v}} - V_h^{\pi_\mathbf{v}}$, where $V_h^{\pi_\mathbf{v}}(s, c) := \sum_{a \in \mathcal{A}} \pi_\mathbf{v}(a|s)P_h^{\pi_\mathbf{v}}(s, a, c)$ for any $(s, c) \in \mathcal{S} \times \mathbb{R}$. Furthermore, the theorem holds holds even if a policy dependent on $C_h$ is used. In our experiments, we indeed used such a policy.*

Based on Theorem 2 and Lagrangian (3), we propose an algorithm called chance-constrained policy gradient (CCPG) shown in Algorithm 1.

## 6 EXPERIMENTS

We verified whether the proposed algorithms can learn constraints and obtain a policy that satisfies the chance constraints using the CCPG algorithm.

**Implementation Details:** For data collection, we used an RL agent trained by PPO (Schulman et al., 2017) to maximize returns without any constraints. We then labeled the data using the true cost functions. For estimating the cost functions, we employed a neural network with three linear layers, with batch normalization and ReLU in the middle layer and a sigmoid function in the output layer. NAdam (Dozat, 2016) was used to update the network. Furthermore, we integrated CCPG into PPO. For all experiments, we set the chance constraint parameter $\delta$ to 0.1. In addition, all experiments were conducted with five different random seeds.

**Baseline:** We used Inverse Constrained Reinforcement Learning (ICRL) (Malik et al., 2021), the most popular method in this field, as a baseline. Since original ICRL updates the policy to satisfy expected constraints by estimated cost functions, we modified it to update the policy to satisfy chance constraints. This means that the cost function is estimated by ICRL, but the policy is optimized by CCPG. Also, in the original ICRL, the cost function was assumed to depend only on the current state and action, but in this study it has been modified to depend on the next state as well.

## 6.1 FROZEN LAKE (TABULAR ENVIRONMENT)

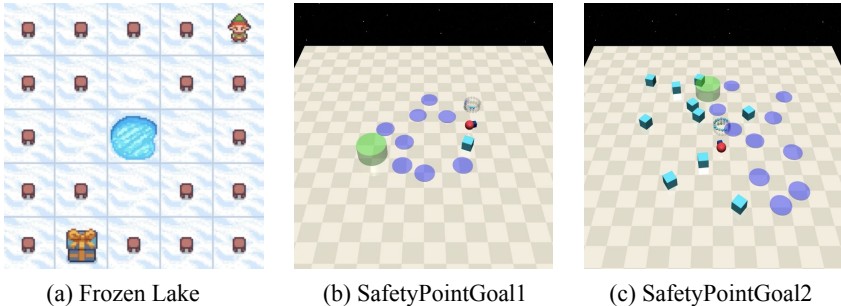

| (a) Frozen Lake | (b) SafetyPointGoal1 | (c) SafetyPointGoal2 |

Figure 2: The environments used in the experiments. (a) In FrozenLake, reaching to the top, bottom, left or right grid of the hole is constraint violation. Brown objects in grids are chairs. (b) In SafetyPointGoal-1, entering the hazard area indicated in blue is constraint violation. (c) In SafetyPointGoal-2, either entering the hazard or colliding with the vase indicated in light blue.

First, we conducted an experiment using modified Frozen Lake (Figure 2 (a)) in OpenAI Gym as a simple tabular environment. It is a 5x5 grid environment in which an agent has to navigate from a randomly set starting point (one of chairs) to a designated goal while satisfying a constraint. The action space is $\{UP, DOWN, LEFT, RIGHT, STAY\}$. If the agent selects $STAY$, it will remain there with a probability of 1. If the agent selects other actions, the agent can go in the selected direction with a probability of $1/3$, and go to the left or right direction of the selected direction with the remaining probability of $2/3$. If the agent falls into the hole in the center, the agent will stay there with probability 1 no matter which action the agent selects. The reward is 1 when the goal is reached, and always 0 otherwise. Also, in order to have a constraint, the upper, lower, left, and right sides of this hole were set as danger zones. The true constraint is that the cost value is set to $0.2$ when the agent reaches these danger zones, and $-0.3$ otherwise. Here, the horizon $H$ is $50$. And the strategy for selecting the best model is as follows. First, if a model that adheres to the constraints has not yet been obtained, the model that adheres to the constraints as much as possible is selected as the best model. Second, if models that adhere to the constraints have been obtained, the model with the highest return among the models that adhere to the constraints is selected as the best model.

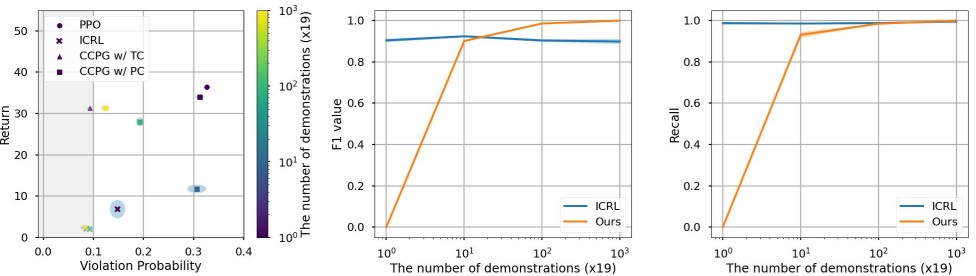

Figure 3: Results in FrozenLake. The left figure shows the relationship between the return and the constraint violation probability. The color of each marker indicates the number of expert demonstration measured in the number of steps, as shown in the colorbar. The shaded area (light gray) indicate the area of constraint compliance. The shaded area surrounding each marker (light blue) represents the standard error. CCPG w/ TC (True Constraints) represents the result of optimizing the policy by CCPG using true constraints. CCPG w/ PC (Predicted Constraints) represents the result of optimizing the policy by CCPG using the constraints predicted by the proposed method. The middle figure represents the relationship between the number of expert data and the F1 value. The right figure represents the relationship between the number of expert data and the recall.

The results of the experiment are shown in Figure 3. The results confirm that the proposed method can recover the exact constraint and learn policies that accurately adhere to the chance constraints when the number of expert demonstration is high. In particular, the proposed method significantly outperforms ICRL and achieves a return of more than 30. Furthermore, as shown in Figure 4, the

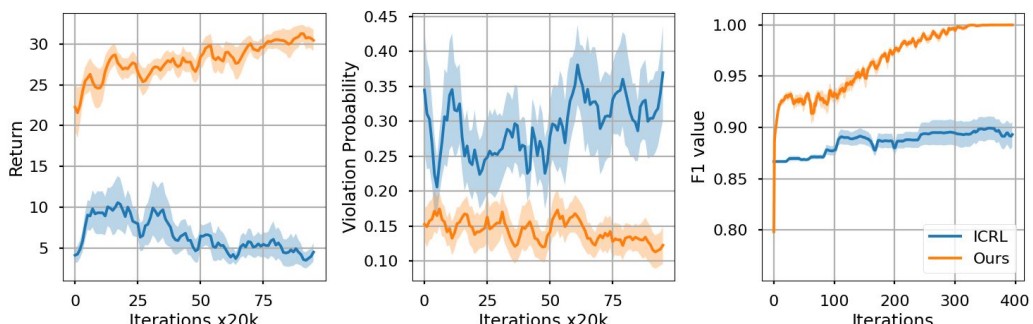

Figure 4: Learning curves for ICRL and the proposed method in Frozen Lake. These values are moving average values with window size = 5.

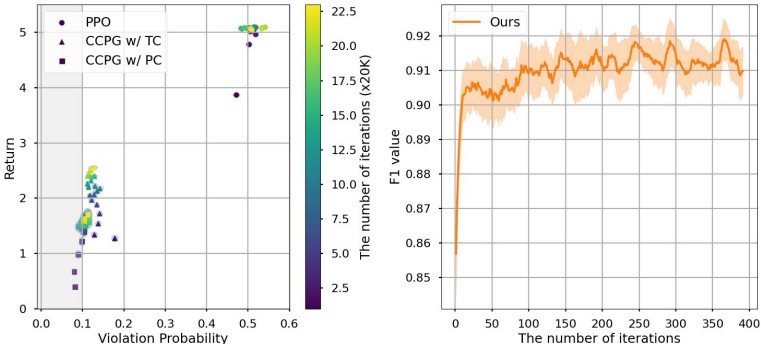

Figure 5: Results in SafetyPointGoal-1. The left figure shows the relationship between the return and the constraint violation probability. The color of each point indicates the number of iterations, with yellow representing high iterations. These values are moving average values with window size = 3. The right figure represents the relationship between the number of iterations and the F1 value. These values are moving average values with window size = 10. Also, the color of the background indicates the standard error.

proposed method also has an advantage in learning stability, which is expected to improve performance consistently.

On the other hand, ICRL performed slightly better in terms of constraint compliance. This can be attributed to the fact that it uses pessimistic estimate of the true constraint, which is confirmed by the fact that the recall value is unduly high in the right in Figure 3. As observed in Gaurav et al. (2023), ICRL and other previous works alternate between cost estimation and policy update, which tends to make the learning process very unstable. As a result, the recall value is unduly high. In addition, because ICRL is trained online, it is able to treat non-demonstration data for cost function estimation, which may be the reason for the higher f1 and recall. In contrast, the proposed method estimates the cost function offline in a stable manner, and it shows superior performance in terms of both stability and return.

## 6.2 SAFETY GYMNASIUM (CONTINUOUS ENVIRONMENT)

To verify whether the proposed method can be used in more complex environments, we conducted experiments using the SafetyPointGoal-1 environment in the Safety Gymnasium (Ji et al., 2023) environment (Figure 2 (b)) where the observation space and action space are continuous. This is an environment in which navigation is performed to a randomly generated goal point while avoiding entering hazards (dangerous areas). Here, to estimate a cost function, 23,000 expert data were used.

The results of the experiment are shown in Figure 5. The results confirm that even in complex environments with continuous action and observation spaces, the proposed method is able to recover the cost function with high accuracy and also obtain the policy to protect the constraints. Actually,

we also performed experiments using the prior work ICRL. However, in ICRL, it is necessary to calculate the weight of importance sampling $\omega(\tau) = \Pi_{h=1}^{H}\omega(s_h, a_h)$, and this value diverged in this experiment with a long horizon $H = 200$. So the policy and the cost function could not be learned. On the other hand, our proposed method is able to learn the cost function stably even when the task is complex and the episode length is long.

## 6.3 MULTI-CONSTRAINTS ESTIMATION

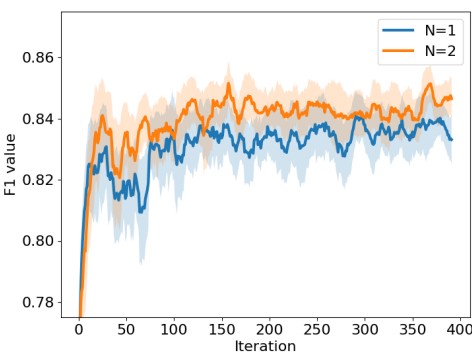

Figure 6: Results in SafetyPointGoal-2 where there are two true constraints. $N = 1$ and $N = 2$ represent the results of estimating the constraints with the number of cost functions as 1 and 2 respectively in the proposed method. These values are moving average values with window size = 10. Also, the color of the background indicates the standard error.

We experimented with the SafetyPointGoal-2 environment (Figure 2 (c)) in SafetyGymnasium to see if multiple cost estimation is possible. In this environment, the agent must reach a goal while avoiding not only hazards but also vases. If the agent encounters hazards for more than 10 steps, it violates the constraint on hazards, and if the agent encounters vases for more than 40 steps, it violates the constraint on vases.

Figure 6 shows the results of the cost function estimation with the number of constraints as $N = 1, 2$. It is shown that in the case of two true constraints, the accuracy of constraint recovery is higher when estimating with $N = 2$ than when estimating with $N = 1$. This result suggests that the proposed decision and loss functions were shown to be potentially useful in estimating multiple constraints.

## 7 RELATED WORK

**Reinforcement Learning from Human Feedback (RLHF):** The most commonly employed methods of giving feedback are to return whether the behavior is absolutely good or bad (Warnell et al., 2018; Arakawa et al., 2018) or to compare two trajectories and choose the superior one (Akrour et al., 2012; Wirth et al., 2016; Christiano et al., 2017; Lee et al., 2021a;b; Liu et al., 2022). In this study, the absolutely good or bad (whether constraints are satisfied or not) is given as a feedback. In recent years, RLHF has been applied to various fields. In meta reinforcement learning, RLHF has been applied to understand common human preferences in multiple tasks (Ding et al., 2023; Joey Hejna and Sadigh, 2023). Ren et al. (2022) utilized human feedback to quickly identify tasks in multi-task RL. It has also been used in research that attempts to create diversity in agent behavior by selecting the two most different from three trajectories (Wang et al., 2023). Furthermore, it is beginning to be used in training Large Language Models (Ouyang et al., 2022; Wu et al., 2023; Dai et al., 2024), operating real robots (Choi et al., 2020; Ding et al., 2023), and in treatment recommendation systems (Xu et al., 2021). On the other hand, in the real world, there are behaviors that we never want the agent to take and that should be restricted. However, it is difficult to restrict these behaviors with reward-based learning methods.

**Constrained Reinforcement Learning:** In ordinary RL, the goal is to obtain a policy that maximizes rewards. On the other hand, there are cases where we want to avoid catastrophic situations at the same time. To address this, constrained RL has been actively studied.

Methods that maximize returns while keeping a certain cost function below a certain boundary have often been studied (Hordijk and Kallenberg, 1984; Geibel, 2006; Kadota et al., 2006; Stooke et al., 2020; Roy et al., 2022). Among them, many methods have been studied that use the method of

Lagrange multiplier to solve the problem (Kadota et al., 2006; Stooke et al., 2020; Roy et al., 2022). While these methods aim to obtain policies that satisfy expected constraints, (Chow et al., 2017; Petsagkourakis et al., 2022; Chen et al., 2023) proposed methods to obtain policies that satisfy chance constraints, which are probabilistic constraints, by using state augmentation or policy gradient methods, etc. In addition, Sootla et al. (2022) proposed a method that satisfies the constraint almost surely by using a simple state augmentation method. We proposed a Lagrangian-based method for solving chance constrained RL.

The above methods assume that constraints are describable and known, which is often not the case in practice. Therefore, methods have been proposed to estimate constraints using data generated by humans or feedback from humans. The most actively studied method is to estimate the cost function from human demonstrations by applying inverse RL techniques (Scobee and Sastry, 2020; Glazier et al., 2021; Papadimitriou and Anwar, 2021; Stocking et al., 2021; Chou et al., 2022; Malik et al., 2021; Gaurav et al., 2023). However, demonstrations are either prohibitively expensive or sometimes impossible to collect. Therefore, instead of estimating constraints from the demonstration data, some methods have been proposed to estimate constraints from physical force (Zhu et al., 2024), trajectory preference (Dai et al., 2024; Chirra et al., 2024; Peng and Billard, 2024) and stop signal (Poletti et al., 2023) from humans. These methods to estimate constraints from human feedback often succeed in recovering constraints, but there is no theoretical guarantee that the optimal policy can be obtained. Although (Lindner et al., 2024) is guaranteed to yield an optimal policy, it is only feasible when the features of the policies are available, which is not realistic. In this study, we propose a method that does not require expensive demonstrations and unrealistic assumptions and is theoretically guaranteed to yield the optimal policy.

## 8 CONCLUSION

In this paper, we showed that there exists a natural correspondence between human feedback and constraint formulation. Based on it, we propose a new decision function for CLHF (from trajectory feedback) that can manage multiple constraints and ambiguity in human feedback. Builsing on those theoretical results, we propose a new policy gradient algorithm for solving constrained RL problems with chance constraints. Finally, we empirically showed superior performance of our algorithms.

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

# A    PROOF OF PROPOSITION 1

**In this proof, we always impose an assumption on feedback generation as explained below.**
That will not lose generality; since imposing assumptions on feedback generation process means
that we are restricting ourselves to easier CLHF cases, hardness for difficult cases obviously follows.
We also fix $H$ and $N$ to 1.

Concretely, we assume that feedback is "clear", meaning that

$$\rho(+1|\tau) = \begin{cases} 1 & \text{if } \nu(\tau) > 0 \\ 0 & \text{if } \nu(\tau) \leq 0 \end{cases}$$

for trajectory feedback, and

$$\rho'(+1|\pi) = \begin{cases} 1 & \text{if } \nu'(\pi) > 0 \\ 0 & \text{if } \nu'(\pi) \leq 0 \end{cases}$$

for policy feedback.

Suppose an MDP $\mathcal{M}$. For any scalar $\xi \in [0, 1)$ and a function $f : \mathcal{S} \times \mathcal{A} \times \mathcal{S} \to [-1, 1]$, let $\Pi$ be a
set of policies, and

$$\Pi_C(f, \xi; \mathcal{M}) := \{\pi | \pi \text{ is safe in a C-CMDP induced by } \mathcal{M}, f, \text{ and } \xi\} \subset \Pi,$$
$$\Pi_E(f; \mathcal{M}) := \{\pi | \pi \text{ is safe in an E-CMDP induced by } \mathcal{M} \text{ and } f\} \subset \Pi.$$

The following proposition is a formal version of Proposition 1 regarding trajectory feedback.

> **Proposition 3.** *There exists an MDP $\mathcal{M}$ and a cost function $c_1$ such that the E-CMDP induced by $\mathcal{M}$
> and $c_1$ has uncountably many $\widehat{c}_1 : \mathcal{S} \times \mathcal{A} \times \mathcal{S} \to [-1, 1]$ satisfying that $\Pi_E(c_1; \mathcal{M}) \subsetneq \Pi_E(\widehat{c}_1; \mathcal{M})$
> although*
> $$\mathbb{I}\left(\sum_{h=1}^{H} \widehat{c}_1(s_h, a_h, s_{h+1}) > 0\right) = \rho(+1|\tau) \tag{4}$$
> *for any trajectory $\tau \in \mathcal{T}$.*

Before its proof, let us explain its implication.

What Equation (4) states is that the estimated cost function perfectly reconstructs $\rho(+1|\cdot)$, and thus,
perfectly fits any trajectory feedback dataset. Nonetheless, from $\Pi_E(c_1; \mathcal{M}) \subsetneq \Pi_E(\widehat{c}_1; \mathcal{M})$, it is
implied that the estimated set of safe policies actually contains an unsafe policy.

*Proof of Proposition 3.* For the proof, we use the following two-state two-action MDP:
- $\mathcal{S} = \{s_1, s_2\}$,
- $\mathcal{A} = \{a_1, a_2\}$,
- $P_1(s_1) = 1$ and $P_1(s_2) = 0$,
- $P(s_1|s_1, a_1) = 1$ and $P(s_2|s_1, a_1) = 0$,
- $P(s_1|s_1, a_2) = (1 - p)/2$ and $P(s_2|s_1, a_2) = (1 + p)/2$,

where $p \in [0, 1)$. We set

$$c_1(s, a, s') = \mathbb{I}(s' = s_2) - \frac{1}{2}\mathbb{I}(s' = s_1).$$

Now, suppose the E-CMDP induced by $\mathcal{M}$ and $c_1$. In this E-CMDP,

$$\Pi_E(c_1; \mathcal{M}) = \left\{\pi \,\middle|\, \pi(a_2|s_1) \leq \frac{2}{3(1 + p)}\right\}.$$

Indeed,

$$v_{c_1}^\pi(P_1)$$
$$= \pi(a_2|s_1)[c_1(s_1,a_2,s_1)P(s_1|s_1,a_2) + c_1(s_1,a_2,s_2)P(s_2|s_1,a_2)] + c_1(s_1,a_1,s_1)\pi(a_1|s_1)$$
$$= \pi(a_2|s_1)\left[P(s_2|s_1,a_2) - \frac{P(s_1|s_1,a_2)}{2}\right] - \frac{1}{2}\pi(a_1|s_1)$$
$$= \pi(a_2|s_1)\left[\frac{1}{2} + P(s_2|s_1,a_2) - \frac{P(s_1|s_1,a_2)}{2}\right] - \frac{1}{2}$$
$$= \frac{3(1+p)}{4}\pi(a_2|s_1) - \frac{1}{2}\,.$$

On the other hand, trajectories are labeled as unsafe if and only if $s_2$ is reached, so any $\widehat{c}_1$ such that

$$\widehat{c}_1(s_1,a_2,s_1) = -1 \text{ and } 0 < \widehat{c}_1(s_1,a_2,s_2) \leq \frac{7-5p}{5(1+p)}$$

satisfies [Equation (4)](#) while

$$v_{\widehat{c}_1}^\pi(P_1)$$
$$= \pi(a_2|s_1)[\widehat{c}_1(s_1,a_2,s_1)P(s_1|s_1,a_2) + \widehat{c}_1(s_1,a_2,s_2)P(s_2|s_1,a_2)] + \widehat{c}_1(s_1,a_1,s_1)\pi(a_1|s_1)$$
$$= \pi(a_2|s_1)[1 + \widehat{c}_1(s_1,a_2,s_2)P(s_2|s_1,a_2) - P(s_1|s_1,a_2)] - 1$$
$$= \frac{(1+p)(1+\widehat{c}_1(s_1,a_2,s_2))}{2}\pi(a_2|s_1) - 1$$
$$\leq \frac{6}{5}\pi(a_2|s_1) - 1\,,$$

which means that

$$\Pi_E(c_1;\mathcal{M}) \subsetneq \left\{\pi \,\middle|\, \pi(a_2|s_1) \leq \frac{5}{6}\right\} \subset \Pi_E(\widehat{c}_1;\mathcal{M})\,.$$

This concludes the proof. $\qquad\square$

The following proposition is a formal version of [Proposition 1](#) regarding policy feedback.

> **Proposition 4.** *For any $\delta \in [0,\infty)$, there exists an MDP $\mathcal{M}$ and a cost function $c_1$ such that the C-CMDP induced by $\mathcal{M}$, $c_1$, and $\delta$ has uncountably many $\widehat{c}_1 : \mathcal{S} \times \mathcal{A} \times \mathcal{S} \to [-1,1]$ satisfying that $\Pi_C(c_1,\delta;\mathcal{M}) \subsetneq \Pi_C(\widehat{c}_1,\delta;\mathcal{M})$ although*
>
> $$\mathbb{I}\bigl(v_{\widehat{c}_1}^\pi(P_1) > 0\bigr) = \rho'(+1|\pi) \qquad (5)$$
>
> *for any policy $\pi \in \Pi$.*

Before its proof, let us explain its implication.

What [Equation (5)](#) states is that the estimated cost function perfectly reconstructs $\rho'(+1|\cdot)$, and thus, perfectly fits any policy feedback dataset. Nonetheless, from $\Pi_C(c_1,\delta;\mathcal{M}) \subsetneq \Pi_C(\widehat{c}_1,\delta;\mathcal{M})$, it is implied that the estimated set of safe policies contains an unsafe policy.

*Proof of [Proposition 4](#).* For the proof, we use the same MDP as the one used in the proof of [Proposition 3](#). We set $\delta = p$, and

$$c_1(s,a,s') = \frac{1-p}{1+p}\mathbb{I}(s' = s_2) - \mathbb{I}(s' = s_1)\,.$$

Now, suppose the C-CMDP induced by $\mathcal{M}$, $c_1$, and $\delta = p$. In this C-CMDP,

$$\Pi_C(c_1,\delta;\mathcal{M}) = \left\{\pi \,\middle|\, \pi(a_2|s_1) \leq \frac{2p}{1+p}\right\}\,.$$

Indeed,

$$\mathbb{P}^{\pi}(\nu(T) \leq 0) = 1 - \pi(a_2|s_1)P(s_2|s_1, a_2)$$
$$= 1 - \pi(a_2|s_1)\frac{1+p}{2}\,.$$

On the other hand, any policy can be verified to be labeled as safe by policy feedback. To see this, note that a policy $\pi$ with $\pi(a_2|s_1) = 1$ has the highest expected cost, which is

$$v_{c_1}^{\pi}(P_1) = c_1(s_1, a_2, s_1)P(s_1|s_1, a_2) + c_1(s_1, a_2, s_2)P(s_2|s_1, a_2) = -\frac{1-p}{2} + \frac{1-p}{2} = 0\,.$$

Therefore, policy feedback does not provide no information on $c_1$. Now, it is not really difficult to find $\widehat{c}_1$ such that $\Pi_C(c_1, \delta; \mathcal{M}) \subsetneq \Pi_C(\widehat{c}_1, \delta; \mathcal{M})$, and we omit the rest of the proof. $\qquad\square$

## B    PROOF OF THEOREM 1

To prove Theorem 1, we need several lemmas.

The following lemma almost immediately follows from Lemma 4 of Russo and Van Roy (2013). Note that we need it to allow the sequential data collection process as in Figure 1 rather than iid data collection process frequently assumed in supervised learning.

**Lemma 1.** *Consider random variables $(Z_k \mid k \in \mathbb{N})$ adapted to the filtration $(\mathcal{F}_k \mid k \in \{0\} \bigcup \mathbb{N})$. Assume that there are random variables $(U_k \mid k \in \mathbb{N})$ that are also adapted to the filtration $(\mathcal{F}_k \mid k \in \{0\} \bigcup \mathbb{N})$ and satisfy $Z_k \leq U_k$ for all $k$ almost surely, and $\mu_k := \mathbb{E}[\exp(\lambda U_k) \mid \mathcal{F}_{k-1}]$ exists and is bounded for all $k$ and non-negative scalar $\lambda$ almost surely. Letting*

$$\psi_k(\lambda) := \log \mathbb{E}[\exp\{\lambda(U_k - \mu_k)\} \mid \mathcal{H}_{k-1}]$$

*be the conditional cumulant generating function, for any non-negative scalars $x$ and $\lambda$,*

$$\mathbb{P}\left(\lambda \sum_{k=1}^{K} Z_k \leq x + \sum_{k=1}^{K} (\psi_k(\lambda) + \lambda\mu_k) \text{ for all } K \in \mathbb{N}\right) \geq 1 - e^{-x}$$

Next, we are going to prove the following lemma.

**Lemma 2.** *For a trajectory $\tau \in \mathcal{T}$, let $f_\tau : [-H, H] \to [0, \infty)$ be a real-valued function defined by $f(x) := \rho(+1|\tau)\ell'(x) - \rho(-1|\tau)\ell'(-x)$. Under Assumption 2, $(d^{\star}(\tau) - x)f_\tau(x) \leq 0$ for any $\tau \in \mathcal{T}$ and $x \in [-H, H]$.*

*Proof.* Recall that for each $\tau \in \mathcal{T}$, the optimal decision function is given by

$$d^{\star}(\tau) \in \underset{x \in [-H, H]}{\arg\min} [\rho(+1|\tau)\ell(x) + \rho(-1|\tau)\ell(-x)]\,.$$

From Assumption 2, $f$ is differentiable and takes a minimum in $[-H, H]$. Therefore, $d^{\star}(\tau)$ is a well-defined minimizer. The claim holds from the first-order necessary condition for optimality. $\qquad\square$

Now, we are ready to prove Theorem 1.

*Proof.* For the time being, consider a fixed $d$. We later take union bound.

From Assumption 2,

$$\Re_{\mathcal{D}}(d) - \Re_{\mathcal{D}}(d^{\star}) = \mathbb{E}_{(T,Y)\sim\mathcal{D}}[\ell(Yd(T)) - \ell(Yd^{\star}(T))]$$
$$\geq \mathbb{E}_{(T,Y)\sim\mathcal{D}}\Big[-Y(d^{\star}(T) - d(T))\ell'(Yd^{\star}(T)) + \sigma(d(T) - d^{\star}(T))^2\Big]\,.$$

Therefore, setting $\mathcal{H}_{k-1}$ to be the $\sigma$-algebra generated by $(\tau_1, y_1, \ldots, \tau_{k-1}, y_{k-1}, \tau_k)$, $Z_k = \ell(y_k d^\star(\tau_k)) - \ell(y_k d(\tau_k))$, and $U_k = y_k(d^\star(\tau_k) - d(\tau_k))\ell'(y_k d^\star(\tau_k)) - \sigma(d(\tau_k) - d^\star(\tau_k))^2$,

$$\mu_k = \mathbb{E}\{y_k(d^\star(\tau_k) - d(\tau_k))\ell'(y_k d^\star(\tau_k)) \mid \mathcal{H}_{k-1}\} - \sigma(d(\tau_k) - d^\star(\tau_k))^2$$

$$\leq -\sigma(d(\tau_k) - d^\star(\tau_k))^2$$

$$\psi_k(\lambda) = \log \mathbb{E}[\exp\{\lambda y_k(d^\star(\tau_k) - d(\tau_k))\ell'(y_k d^\star(\tau_k))\} \mid \mathcal{H}_{k-1}]$$

$$\leq \frac{\lambda^2 L^2 (d^\star(\tau_k) - d(\tau_k))^2}{2},$$

where the upper bound of $\mu_k$ follows from Lemma 2, and the upper bound of $\psi_k$ follows from Hoeffding's inequality (Lemma A.1 in (Cesa-Bianchi and Lugosi, 2006)) together with the Lipschitz continuity of $\ell$, which implies that $\ell'$ is bounded by $L$. Applying Lemma 1, we deduce that

$$\mathbb{P}\left(\forall K \in \mathbb{N}, \mathfrak{R}_\mathcal{D}(d^\star) - \mathfrak{R}_\mathcal{D}(d) \leq \frac{x}{\lambda K} + \left(\frac{\lambda L^2}{2K} - \frac{\sigma}{K}\right)\sum_{k=1}^K (d(\tau_k) - d^\star(\tau_k))^2\right) \geq 1 - e^{-x}.$$

Rearranging and choosing $\lambda = \sigma/L^2$, and $x = \log(1/\delta')$,

$$\mathbb{P}\left(\forall K \in \mathbb{N}, \mathfrak{R}_\mathcal{D}(d^\star) - \mathfrak{R}_\mathcal{D}(d) \leq \frac{L^2}{\sigma K} \log\frac{1}{\delta'} - \frac{\sigma}{2K}\sum_{k=1}^K (d(\tau_k) - d^\star(\tau_k))^2\right) \geq 1 - \delta'.$$

As it holds only for a fixed $d$, we are going to take union bound below.

Now, let $\mathcal{H}_\alpha$ be an $\alpha$-cover of $\mathcal{H}$ with $\|\cdot\|_\infty$. Furthermore, let $d_\alpha \in \mathcal{H}_\alpha$ be a function such that $\|d - d_\alpha\|_\infty \leq \alpha$. Then, since $\ell$ is $L$-Lipschitz,

$$\mathfrak{R}_\mathcal{D}(d_\alpha) + \alpha L \geq \mathfrak{R}_\mathcal{D}(d).$$

Accordingly, setting $\delta' = \delta/\mathcal{N}_\alpha$,

$$\mathbb{P}\left(\forall (d, K) \in \mathcal{H} \times \mathbb{N}, \mathfrak{R}_\mathcal{D}(d^\star) - \mathfrak{R}_\mathcal{D}(d) \leq \alpha L + \frac{L^2}{\sigma K} \log\frac{\mathcal{N}_\alpha}{\delta} - \frac{\sigma L_{2,K}(d, d^\star)}{2}\right) \geq 1 - \delta.$$

This concludes the proof. $\qquad\square$

## C  PROOF OF THEOREM 2

*Proof of Theorem 2.* Let $p_\mathbf{v}(\tau)$ be the probability of $\tau$ under a policy $\pi_\mathbf{v}$. Then,

$$\nabla_\mathbf{v} \mathbb{P}^{\pi_\mathbf{v}}\left[\max_{n \in [N]} C_{n,H+1} > 0\right] = \sum_{\tau \in \mathcal{T}} \mathbb{I}\left(\max_{n \in [N]} C_{n,H+1} > 0\right)\nabla_\mathbf{v} p_\mathbf{v}(\tau)$$

$$= \mathbb{E}^{\pi_\mathbf{v}}\left[\mathbb{I}\left(\max_{n \in [N]} C_{n,H+1} > 0\right)\nabla_\mathbf{v} \ln p_\mathbf{v}(\tau)\right].$$

Since $p_\mathbf{v}(\tau) = P_1(s_1)\prod_{h=1}^H P(s_{h+1}|s_h, a_h)\pi_\mathbf{v}(a_h|s_h)$,

$$\nabla_\mathbf{v} \mathbb{P}^{\pi_\mathbf{v}}\left[\max_{n \in [N]} C_{n,H+1} > 0\right] = \sum_{h=1}^H \mathbb{E}^{\pi_\mathbf{v}}\left[\mathbb{I}\left(\max_{n \in [N]} C_{n,H+1} > 0\right)\nabla_\mathbf{v} \ln \pi_\mathbf{v}(A_h|S_h)\right].$$

By the law of total expectation,

$$\mathbb{E}^{\pi_\mathbf{v}}\left[\mathbb{I}\left(\max_{n \in [N]} C_{n,H+1} > 0\right)\nabla_\mathbf{v} \ln \pi_\mathbf{v}(A_h|S_h)\right] = \mathbb{E}^{\pi_\mathbf{v}}[P_h^{\pi_\mathbf{v}}(S_h, A_h, C_h)\nabla_\mathbf{v} \ln \pi_\mathbf{v}(A_h|S_h)].$$

This concludes the proof. $\qquad\square$

