# OpenReview forum: "Foundation of Scalable Constraint Learning from Human Feedback"
_ICLR.cc/2025/Conference — ICLR 2025 Conference Withdrawn Submission_

### Official Review · Reviewer_iPaU · 2024-10-30

**Soundness:** 2
**Presentation:** 1
**Contribution:** 2
**Rating:** 3
**Confidence:** 4

**Summary:**

The paper focuses on constraint learning from human feedback (CLHF) and proposes a new decision function based on trajectory feedback for CLHF. It demonstrates that cost functions learned from feedback may not accurately represent the true constraints, which can result in policies that appear safe but are not actually safe in the true environment. The authors also provide some theoretical analysis to support their claims.

**Strengths:**

1) CLHF is a valuable research direction and it is important for both reinforcement learning applications and the security of large language models (LLMs).
2) The authors propose a new decision function based on trajectory feedback and provide the theoretical analysis for the proposed decision function.

**Weaknesses:**

1) There is no clear representation of the motivation and contributions of the paper.
2) The theoretical analysis is limited, aspects such as convergence analysis and constraint violations related to constraint learning are not addressed.
3) The overall presentation of the paper might benefit from improvements, as it does not clearly convey its main claims and contains some expression errors.
4) The experimental results do not seem to adequately support the theoretical analysis. For example, Figure 3 shows significant constraint violations in CCPG w/PC.
5) The paper lacks additional necessary experiments, including comparison experiments, ablation studies, and hyperparameter analysis, etc.
6) The baselines and experimental environments in the paper are too few to illustrate the validity of the method.
7) There is no code or detailed implementation description provided to support the reproducibility of the results.

**Questions:**

1) What do $q$ and $P$ represent in Algorithm 1?
2) Could the authors discuss the reasons behind the constraint violations observed in CCPG during the experiments, as well as the differences in performance between CCPG w/PC and CCPG w/TC across different environments?

---

### Official Review · Reviewer_uEdH · 2024-11-01

**Soundness:** 3
**Presentation:** 2
**Contribution:** 3
**Rating:** 5
**Confidence:** 3

**Summary:**

This paper provides a theoretical analysis on the feasibility of constraint learning from two types of human feedback and provides results quantifying the learning complexity of multiple constraint learning from trajectory-based feedback. The authors furthermore introduce a policy-gradient formulation based on the lagrangian objective corresponding to the latter theoretical result and develop a practical algorithm, labelled CCPG, that jointly learns constraints and a policy from trajectory-labelled data.

**Strengths:**

The paper addresses a highly relevant problem, namely the inference of constrained objectives from coarsely-labelled data and its theoretical foundations:
- The authors provide a thorough theoretical analysis of this learning paradigm in the case of trajectory feedback and policy feedback and show (to my knowledge) novel insights regarding the feasibility of constraint learning depending on the type of feedback.
- The main theoretical finding establishes a risk bound that quantifies the learning complexity of learning constraints in terms of (among others) trajectory count, model class and permitted constraint violation probability.

These results are relevant to the field, as they lay a sound motivation for practical algorithms. The empirical results are furthermore reasonably aligned with the theoretical analysis.

**Weaknesses:**

- The proof of proposition 1 leaves some questions open to me (see questions).
- I believe, a natural addition that many readers would appreciate in this context is if the analysis extended to preference-based feedback.
- In addition to the point above, it would be interesting to see an experimental confirmation of the contents of proposition 1 and 2 (i.e. the learnability of constraints from diffferen types of feedback).


Generally speaking, the paper is written in concise and clear language, however there are a number of typos and sentences/notation I would consider rephrasing:
- line 39, "comprehensive" is too strong in my view
- line 40, "fills"
- line 51, "a more general constraints"
- line 53, "empirical validated"
- line 57, I don't know the term "curly alphabets"
- line 65, "finte"
- line 79, the shorthand for the value function taking a probability distribution as an input should be mentioned
- line 110, "a constrained RL"

**Questions:**

Concerning Propositions 1 & 2:

In proposition 3 and the subsequent paragraph (l. 738) it is stated that an estimated cost function can perfectly reconstruct a trajectory feedback dataset in the sense of Eq. 4) but it is not clear to me why we evalute the estimated cost function $\hat{c}$ through the additional indicator function. In this example MDP, from what I can see a reasonable estimator $\hat{c}_1$ should be able to converge to $\hat{c}_1(s_1,a_2,s_2)=1$ and $0$ for any other possible trajectory. Would this estimate not enable one to recover safe policies?

Concering the impossibility results with policy feedback:

I fail to see the difference between the trajectory feedback setting and a policy feedback setting where annotators observe one trajectory sampled from policy $\pi_k$. And consequently how could one obtain an impossibility result for one, if both feedback forms can be equal?

I believe the overall contribution to be valuable to the literature and am willing to raise my score if the above points are addressed.

---

### Official Review · Reviewer_kf9H · 2024-11-03

**Soundness:** 3
**Presentation:** 1
**Contribution:** 3
**Rating:** 3
**Confidence:** 3

**Summary:**

This paper critically analyzes two forms of human feedback for
constraint inference; namely *trajectory feedback* wherein humans
classify whether given trajectories are safe, and *policy feedback*,
wherein humans classify whether decision policies are safe. Notably, the
paper also considers two forms of constrained MDPs (chance-constrained
MDPs and expected constrained MDPs), and analyzes which forms of human
feedback can induce constraint inference enabling solutions to each type
of constrained MDP. Finally, the paper presents a new policy gradient
method for joint constraint inference and control, and demonstrates its
effectivess at safe policy optimization in some common benchmark tasks.

**Strengths:**

The problem setting studied in this paper is interesting and relevant.
The notion of characterizing which forms of human feedback enable which
types of safe RL is very neat, and so are the theoretical results
towards this end.

**Weaknesses:**

One major weakness of the paper is that it is not well-written; examples
are given below and in the Questions section. As a consequence, I found
it fairly difficult to follow the paper—I had to write my own document
of notes and math to fill in several missing or confusing pieces. The
clarify of the paper could have been improved tremendously, and I
encourage the authors to do so.

The part of the proof of Proposition 4 that you omitted should not have
been omitted, and this proof can be explained better. The claims on ine
816-820 are confusing. Particularly, it says "policy feedback does not
provide no information on $c_1$". Firstly, it should say "policy
feedback does not provide any information…". More importantly, I don't
necessarily find this accurate. What the equation on line 818 implies is
that for any policy feedback dataset will have the form
$\{(\pi_k, 0)\}$, so there is no signal for identifying a safe policy.
This property is directly a consequence of the stucture of $c_1$, though
(i.e., there are certainly alternative cost functions that will not have
this property here). Specifically, it is saying that the costs are
sufficiently close to $0$ (or "symmetric" enough) that no policy is
dangerous. This does give information about $c_1$. However, it does not
prevent us from inferring that $\hat{c}_1\equiv 0$, so that the
constraint function induced from policy feedback does not restrict the
class of safe policies, which can result in the deployment of a
C-CMDP-unsafe policy (by being a strict superset of the set on line
809). I think this example / narrative would be very helpful.

Again, Proposition 2 should have a more complete proof, and perhaps
should be motivated more. What is the oracle here, and why should we
assume access to one? I'm assuming what is really meant here is that
with unbounded preferences, we can learn safe policies. I think there is
a substantial amount missing in the proof here:

1.  The policy feedback case is not immediately obvious to me. Maybe
    it's obvious to you as the authors that have been thinking a lot
    about this setting, but as a reader, I would like verification that
    my thought process is actually what you had in mind; at the very
    least, this would help me understand the claim of the proposition.
    Am I correct in my interpretation that, in the policy feedback case,
    you can theoretically enumerate policies, query the oracle until you
    get feedback $0$, and then this certifies a safe policy by
    definition (so you can return it)?
2.  For the trajectory feedback case, is my interpretation correct that
    you again enumerate policies like in the policy feedback case,
    enumerate trajectories under each policy (given knowledge of
    $P_1, P$), and return a policy once you find one that the oracle
    only returns $1$ on at most $p|\mathcal{T}|$ trajectories? And why
    do you need knowledge of $r$?

Another issue with the paper is what I believe to be a lack of baselines
with respect to the empirical analysis. Notably, the authors had no
results for any baselines except for in the tabular domain. As such,
given the prevalence of the SafetyGym domain, I am skeptical that no
other method is able to solve the tasks attempted in this paper.

## Minor Issues

On line 56, "We denote sets by curly alphabets" – I don't think
"alphabets" is the right word, maybe "braces"?

In Definition 1, "finte" should be "finite".

The notation for the return in Definition 2 is not quite right. In
particular, you're simultaneously defining the return as a function on
$\mathcal{S}$ and on $\Delta(\mathcal{S})$.

The notation $v^\pi_1$ in Definition 3 is technically not defined. I'm
assuming this implicitly represents $v^\pi_{r, h}$ where $r$ is the
reward function in the MDP.

In theorem 1, you have defined $K := |\mathcal{D}|$, so you can (and
should) replace the $\forall (d, K)\in\mathcal{H}\times\mathbb{N}$ with
$\forall d\in\mathcal{H}$.

**Questions:**

Is assumption 1 actually mild? My interpretation of $\nu(\tau) = 0$ is
that no constraints are violated. Assumption 1 says any such policy has
the same preference distribution, which seems bizarre. Consider for
example a CartPole swingup domain where the constraint function is
positive when the velocity of the cart is above a threshold, and zero
otherwise. Then if the system is initially stationary (pole in the
downward position), there is no preference between an agent that does
nothing and one that swings up the pole successfully.

What is $\xi$ supposed to represent on line 722? Is this the constraint
violation probability written as $\delta$ in the main text?

In MDP in the proof of proposition 3, is $H=1$?

In propositions 3 and 4, I think you want probabilities instead of
indicators.

I'm having difficulty following the proof of Proposition 3. How did you
come up with that upper bound on $\hat{c_1}(s_1, a_2, s_2)$? If you're
just going for existence, can't you find much simpler settings of
$\hat{c}$?

In Proposition 4, what is the purpose of the indicator? The clause
inside the indicator is deterministic.

If Lemma 1 is not identical to Russo and Van Roy's Lemma 4, it should
have a proof. What is actually the difference between these two? It
looks to me like you're using $\psi_k$ to be the cumulant generating
function of $U_k$ instead of $Z_k$, is that it? Anyway, I think there
should be a proof here.

Regarding Figure 5, is it known that existing ICRL methods struggle in
this environment? Are there no baselines that can solve this
environment? That sounds suspicious. It also very much weakens the claim
that your method achieves superior performance—I suspect some strong
baselines are missing.

---

### Official Review · Reviewer_ja5U · 2024-11-04

**Soundness:** 2
**Presentation:** 3
**Contribution:** 2
**Rating:** 5
**Confidence:** 4

**Summary:**

This paper aims to provide theoretical results in the context of Constraint Learning from Human Feedback. There are a few impossibility results that have been provided on which type of constraints (expected cost constraint, chance constraint) can be learnt in the context of which models (E-CMDP, C-CMDP).

**Strengths:**

1. The paper is well written and mostly easy to understand.
2. The results about where constraint functions can be learnt or not is quite interesting.
3. The experimental results are quite detailed.

**Weaknesses:**

1. The notion of impossibility in learning is not explained. Does it imply you can never find a cost function that is safe?
2.  There are other types of CMDPs that have not been referenced. Only expected cost constraint and value at risk cost constraint have been provided. There is hard constraint and Conditional Value at Risk constraint that have previously been introduced by Hao et al. [Reward Penalties on Augmented States for Solving Richly Constrained RL Effectively].
3. Why is learning of cost function important to safety? Can't systems be made safer without explicitly learning a cost function
4. How are the experimental domains decided? Only a few environments from Safety Gym are considered. Frozen lake is a simpler problem setting. Why were other environments not considered.
5. In the second paragraph of introduction, it is mentioned that CLHF has significant works. However, in the experiments there are not too many baselines provided. Can I please check why that is the case?
6. There needs to be more intuition provided before utilising formal description. Otherwise, it gets difficult to keep track of all the symbols.

**Questions:**

Please refer to the weaknesses part for the questions. Here are a few more questions:

1. There are other methods in the literature that have been provided for solving C-CMDPs (e.g., Hao et al, Chow et al -- risk constrained RL with percentile risk criteria), so why have the authors introduced a new approach and

2.In 4.1, why are the weights "w" the same for RHS and LHS.

---

### Note · Authors · 2024-11-25

I have read and agree with the venue's withdrawal policy on behalf of myself and my co-authors.